# TrackletMapper: Ground Surface Segmentation and Mapping from Traffic Participant Trajectories

**Jannik Zürn**[1*]  **Sebastian Weber**[1*]  **Wolfram Burgard**[2]

[1]University of Freiburg
[2]University of Technology Nuremberg
zuern@cs.uni-freiburg.de, wolfram.burgard@utn.de

**Abstract:** Robustly classifying ground infrastructure such as roads and street crossings is an essential task for mobile robots operating alongside pedestrians. While many semantic segmentation datasets are available for autonomous vehicles, models trained on such datasets exhibit a large domain gap when deployed on robots operating in pedestrian spaces. Manually annotating images recorded from pedestrian viewpoints is both expensive and time-consuming. To overcome this challenge, we propose *TrackletMapper*, a framework for annotating ground surface types such as sidewalks, roads, and street crossings from object tracklets without requiring human-annotated data. To this end, we project the robot ego-trajectory and the paths of other traffic participants into the ego-view camera images, creating sparse semantic annotations for multiple types of ground surfaces from which a ground segmentation model can be trained. We further show that the model can be self-distilled for additional performance benefits by aggregating a ground surface map and projecting it into the camera images, creating a denser set of training annotations compared to the sparse tracklet annotations. We qualitatively and quantitatively attest our findings on a novel large-scale dataset for mobile robots operating in pedestrian areas. Code and dataset will be made available at http://trackletmapper.cs.uni-freiburg.de.

**Keywords:** Knowledge Distillation, Semantic Segmentation, Navigation

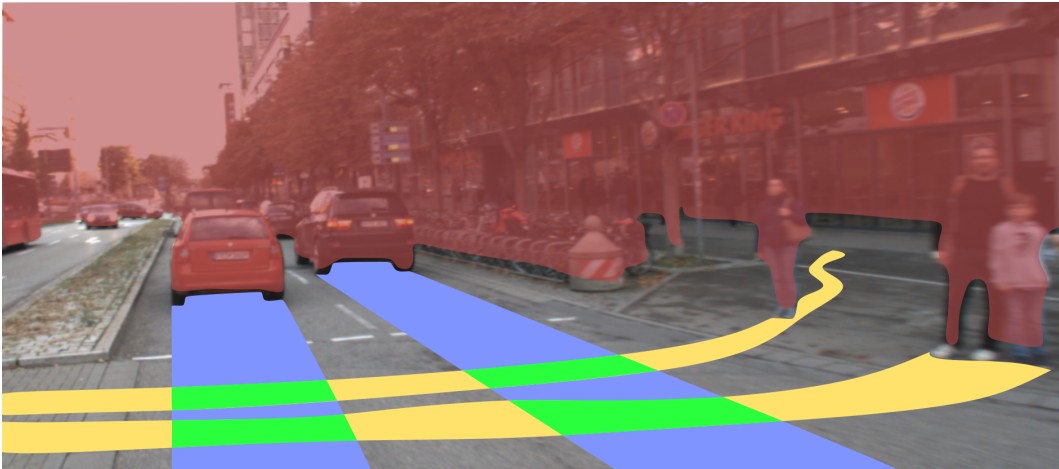

Figure 1: We present *TrackletMapper*, a novel approach for ground surface segmentation that leverages observed traffic participant tracklets to supervise a surface segmentation model. Our pipeline automatically annotates obstacles (red), pedestrian areas/sidewalks (yellow), street crossings (green) and roads (blue) based on the observed trajectories of pedestrians and vehicles.

---

*Equal contribution

6th Conference on Robot Learning (CoRL 2022), Auckland, New Zealand.

# 1 Introduction

Mobile ground robots operating in urban areas encounter a wide range of environments. It is essential for autonomous robots to robustly navigate through such environments even without access to human-annotated map data. Urban environments feature different types of ground surfaces restricted for use only by particular traffic participants. While vehicles are mostly permitted to operate on roads, pedestrians are generally only allowed on sidewalks and in pedestrian areas. Street crossings are permitted to be crossed both by vehicles and by pedestrians. To allow for robust and safe navigation, autonomously operating robots in urban environments are required to localize nearby traffic participants accurately [1, 2, 3] and classify ground surfaces robustly. While autonomous vehicles typically require a binary distinction between road and non-road surfaces, mobile robots operating in pedestrian spaces must crucially be able to distinguish between sidewalks, roads, and road crossings in order to navigate urban environments safely [4, 5, 6].

In recent years, a multitude of semantic segmentation datasets for urban autonomous driving has been proposed [7, 8, 9]. These datasets, however, are recorded from the vantage point of street vehicles. Therefore, models trained on these datasets exhibit a strong bias toward the camera viewpoint, introducing a significant domain gap when deployed in areas intended for non-vehicle usages such as pedestrian areas or sidewalks. While it is possible to manually annotate images obtained from the pedestrian viewpoint, this is an expensive and time-consuming task. Automatic annotation of images offers a promising alternative to manual annotations made by human annotators. Previously proposed automatic annotation approaches typically leverage the ego-motion of a data collection platform to obtain spatially sparse image-level labels of traversable ground surfaces [10, 11] or are based on proprioceptive sensors such as sound and vibration [12, 13, 14]. In contrast to existing work, we additionally leverage the trajectories of other traffic participants such as vehicles and pedestrians, and project them into the camera images. This enables us to label multiple types of ground surfaces, including roads, sidewalks or pedestrian areas, and street crossings based on the type of tracked objects. Hereby, we leverage the fact that under most circumstances, pedestrians walk in areas reserved for them and vehicles drive on roads or through street crossings but not on the sidewalk. The object detector used to generate the pedestrian- and vehicle trajectories does not suffer from the viewpoint-induced domain gap present in segmentation models. To further boost model performance, we build a ground surface map from these predictions by spatially aggregating the predictions. Aggregation of semantic segmentation predictions has been previously proposed [15], however, the generated maps have previously not been used as an annotation source for semantic segmentation models. We show that it is possible to self-distill the segmentation model by re-projecting the aggregated surface map back into the camera images and using them as annotations, boosting the model performance.

In summary, this work offers the following key contributions: (i) A novel automatic annotation approach that leverages trajectories of traffic participants such as vehicles and pedestrians for generating sparse multi-class semantic pixel annotations. (ii) A segmentation model self-distillation pipeline to generate training annotations from projections of an aggregated surface map. (iii) The *Freiburg Pedestrian Scenes* dataset recorded with a robot platform navigating through a wide range of urban pedestrian environments.

# 2 Related Works

Self-supervised methods for visual terrain segmentation in off-road driving applications were investigated in [12, 16, 17, 14, 13]. In these works, labels obtained from a proprioceptive sensor modality (i.e. vibration, sound) are used to partially annotate exteroceptive sensor modalities (i.e. RGB vision). Other non-learning approaches leverage geometric features in LiDAR point clouds to classify vertical and horizontal surfaces [18, 19, 20].

One of the first works to consider auto-generated annotations for semantic image segmentation in the context of autonomous driving was Barnes *et al.* [10]. The authors propose a self-supervised approach for generating drivable paths in monocular RGB images from projected ego-trajectories of the recording vehicle on popular urban driving datasets. Mayr *et al.* [21] propose a self-labeling pipeline for drivable road area segmentation. Based on stereo disparity maps and ground plane fitting, they extract drivable road areas from images and use the annotated RGB images to train a binary segmentation model. Cho *et al.* [22] estimate drivable space and surface normal vectors from stereo images, which are used as pseudo-ground-truth to train a segmentation model. Bruls *et al.* [23]

leverage weakly-labeled annotations for urban road markings based on LiDAR reflectance values and potentials from a Conditional Random Field. Wang *et al.* [24] propose a self-supervised drivable area and road anomaly segmentation approach from RGB-D data. They leverage a stereo depth image to obtain weak labels for obstacles sticking out from the ground level. Wellhausen *et al.* [14] propose a self-supervised weak image labeling scheme based on a proprioceptive vibration-based terrain classifier. Labels predicted by the proprioceptive classifier are projected into the robot camera ego-view. Zürn *et al.* [13] propose a self-supervised labeling scheme based on an unsupervised audio clustering approach, where the cluster indices serve as weak labels and are projected into the robot camera images. Most recently, Onozuka *et al.* [11] propose a traversable area segmentation approach for personal mobility systems such as intelligent wheelchairs.

To summarize, existing methods for automatic annotation or self-supervised approaches do not leverage the additional data provided by the trajectories of other traffic participants, thus, ignoring relevant information. In addition, our work makes use of the aggregated surface map as an additional annotation source, further boosting the segmentation model performance by increasing the number of annotated pixels.

## 3 Technical Approach

Our goal is to label the surface classes *Pedestrian*, *Road*, *Crossing*, and *Obstacle*. The classes *Pedestrian* and *Road* contain surface areas, where either of the two classes is exclusively permitted. Areas intended for pedestrian use include sidewalks, pedestrian zones, and footpaths while vehicle areas include all road sections without crossings. The class *Crossing* is intended to annotate asphalt surfaces at street crossings (zebra crossings or signaled pedestrian crossings). Both pedestrians and vehicles are permitted to cross these areas. Pixels labeled from the ego-trajectory and those obtained from pedestrian trajectories are jointly used to provide annotations for the class *Pedestrian* since we assume that the robot is teleoperated to only traverse pedestrian surfaces. The class *Obstacle* annotates different kinds of non-traversable surfaces such as buildings, moving or static objects extending over ground or vegetation. The class *Unknown* serves as a filler class for all pixels where no annotation is provided. In the following, we will first discuss the automatic generation of image annotations from the robot ego-trajectory and traffic participant tracklets (Subsec. 3.1) and subsequently the generation and projection of the semantic surface map for additional model performance gains (Subsec. 3.2).

### 3.1 Surface Annotations from Tracklets

We first perform LiDAR-SLAM [25], generating a list of poses $\mathbf{p}_i \in \mathbb{SE}(3)$ for the robot base for each data collection run. In the following, we will discuss the projection of the ego-trajectory into image coordinates. In order to project the robot trajectory into the viewpoint of the onboard camera, we associate a time-synchronized robot pose with each of the camera images. Assuming a static transform $\mathbf{T}_C^B$ between the robot base and the camera mounting position relative to the base, the robot trajectory in homogeneous pixel coordinates $\mathbf{u} = [u, v, 1]^T$ can be expressed as

$$\mathbf{u} = \mathbf{K}\mathbf{T}_C^B\mathbf{T}_B^W\hat{\mathbf{p}}, \tag{1}$$

where $\mathbf{T}_B^W$ is the time-dependent transformation between the world coordinates and the current robot base position, obtained from $\mathbf{p}_i$, $\mathbf{K} \in \mathbb{R}^{3 \times 3}$ denotes the intrinsic camera matrix, and $\hat{\mathbf{p}} \in \mathbb{R}^3$ denotes the ego-trajectory projected onto the ground surface. For brevity, we omit the superscript $t$ for time-dependent variables. Note that we dilate the robot trajectory laterally by half its base width in order to label all pixels within the robot footprint.

To obtain the trajectories of other traffic participants such as vehicles and pedestrians, we leverage the ByteTrack [26] object tracker with an EfficientDet [27] object detector pre-trained on the MS-COCO dataset [28]. The object trajectory in 3D world coordinates is obtained by projecting the tracklet bounding box center point coordinates into 3D world coordinates. To perform this transformation, we interpolate sparse depth images obtained from the LiDAR points, which provides an accurate depth estimation for a given object bounding box. Formally, the projection of tracklets into 3D world coordinates $\mathbf{x} \in \mathbb{R}^3$ follows the inverse projection equation:

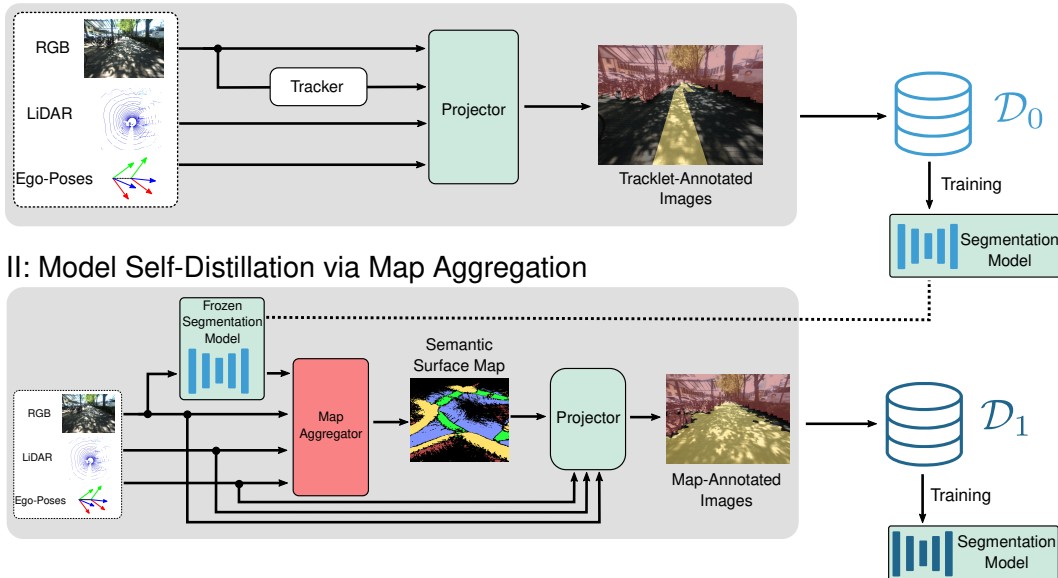

Figure 2: Visualization of our automatic annotation pipeline. In step I, we leverage RGB images, LiDAR point clouds, ego-poses, and an object tracker to project the ego-poses and the observed tracklets into camera images, generating a sparsely-annotated semantic segmentation dataset $\mathcal{D}_0$. In step II, we use a frozen segmentation model trained on $\mathcal{D}_0$ to obtain semantic annotations and aggregate them into a global semantic surface map. Finally, we project this map into the camera images and obtain denser and more consistent annotations, denoted as $\mathcal{D}_1$.

$$\mathbf{x} = \mathbf{T}_W^B \mathbf{T}_B^C d\, \mathbf{K}^{-1}\mathbf{u}, \tag{2}$$

where we follow the same naming convention as in Eq. 1 and the scalar $d \in \mathbb{R}$ denotes the depth scaling factor. Similar to the projection of the ego-trajectory, we assign labels to image pixels according to the 3-D world tracklet projection into image coordinates, according to Eq. 1. Similar to the ego-trajectory, we laterally dilate the tracklet line segments by a fixed object width, which is set to be $0.5\,\mathrm{m}$ for pedestrians and $2\,\mathrm{m}$ for vehicles. Street crossings are defined to be traversable by both pedestrian and motorized traffic participants. We, therefore, define the set of all pixels indicating a street crossing $\mathcal{S}_C$ as the intersection of pixels indicating pedestrian usage $\mathcal{S}_P$ and vehicle usage $\mathcal{S}_V$. More formally, we define $\mathcal{S}_C := \mathcal{S}_P \cap \mathcal{S}_V$. Obstacles are defined as objects extending substantially above the ground plane. To detect the ground plane, we segment the LiDAR point cloud using the pre-trained ground plane estimation network GroundNet [29]. After projecting the segmented point cloud into each RGB image, we label each RGB image pixel located more than $20\,\mathrm{cm}$ above the ground plane as *Obstacle*, following existing stixel-based approaches [30]. We denote the set of so-produced annotations for the surface classes *Pedestrian*, *Road*, *Crossing*, and *Obstacle* as dataset $\mathcal{D}_0$.

### 3.2 Surface Mapping and Self-Distillation via Aggregation and Reprojection

In addition to the aforementioned annotation procedure, we propose a novel self-distillation method for the semantic segmentation model. We argue that the inherent class prediction uncertainty in the segmentation model can be reduced by aggregating multiple predictions for a given patch of ground and re-training the model with these aggregated predictions. Prior works [31, 32] have shown how model self-distillation can help improve model performance. In this work we perform model self-distillation by spatially aggregating predictions in order to re-train the model on these aggregated predictions. Consider a surface patch $S^i$. Following similar formulations by [33] and [34], we associate a belief $\mathbf{h}_t^i \in \mathbb{R}^K$ with $S^i$, containing the log odds vector of $S^i$ being of class $k$. We denote $K$ as the total number of considered classes. We collect all model predictions $\mathbf{p}_i \in \mathbb{R}^K$ that

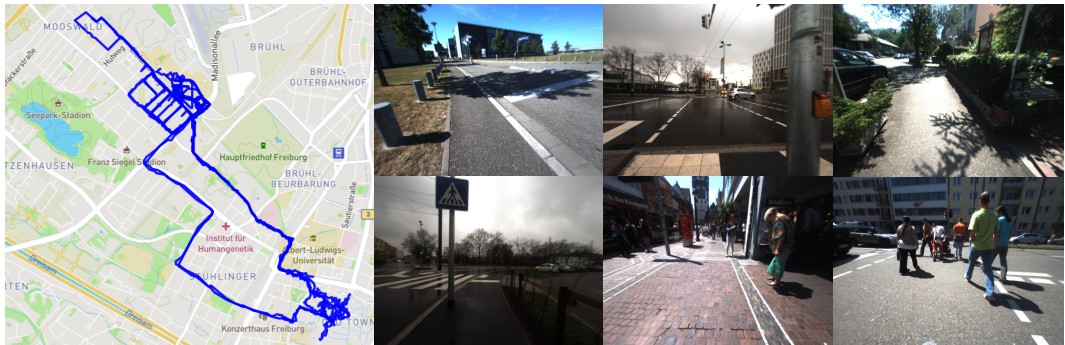

Figure 3: Ego-trajectories (left, blue color) and camera images from our *Freiburg Pedestrian Scenes* dataset. The dataset features a wide range of urban environments including busy streets, pedestrian areas, and road crossings with varying weather conditions.

contain that patch of ground. In the beginning, the vector is initialized with a uniform distribution $\mathbf{h}_0$ over the classes and is updated according to the update rule $\mathbf{h}_{t+1}^i = \mathbf{h}_t^i + (\mathbf{l}_t^i - \mathbf{h}_0)$, where $\mathbf{l}_t^i$ denotes the inverse observation model log odds:

$$\mathbf{l}_t^i = \left[ \log \frac{p_t^i(k=1)}{1 - p_t^i(k=1)}, \quad \log \frac{p_t^i(k=2)}{1 - p_t^i(k=2)}, \quad \cdots, \quad \log \frac{p_t^i(k=K)}{1 - p_t^i(k=K)} \right]^T, \qquad (3)$$

and $p_t^i(k)$ denotes the model prediction for ground surface patch $S^i$ at time step $t$. After all belief updates have been executed, we transform the log-odds vector into class probabilities using the `softmax` function. We take the $\arg\max$ over the probability vector to obtain the most likely surface class and annotate patch $S^i$ with that class. In order to obtain a dense surface representation suitable for training a segmentation model, we triangulate all surface patch center points and create a triangular mesh of ground surfaces. As a post-processing step, we smooth the surface mesh using the Taubin filter [35]. To generate training data for the segmentation model, we again use the 3D poses of the camera and project the semantic surface mesh back into the camera RGB images as dense semantic annotations. Due to the larger spatial extent of the surface map compared to the tracklets, we can significantly increase the number of annotated pixels in each image. We denote the set of so-produced annotations as dataset $\mathcal{D}_1$.

### 3.3 Model Training

The aforementioned annotation scheme labels pixels that are associated with obstacles or have been traversed either by the robot or by other traffic participants. All other pixels in the images are assigned the label *Unknown*. We pose the ground segmentation task as a segmentation task with sparse label supervision, where only a subset of the pixels in each image has annotations available. As the model architecture, we use the DeepLabv3+ model architecture [36]. We use a standard cross-entropy loss for all non-*Unknown* image pixels. *Unknown* pixels are ignored during training.

## 4 Dataset

We present the *Freiburg Pedestrian Scenes* dataset recorded with our robot platform. During each data collection run, the robot is teleoperated through semi-structured urban environments and moves alongside pedestrians on sidewalks, pedestrian areas, and street crossings. Each data collection run consists of time-synchronized sensor measurements from a Bumblebee Stereo RGB camera, a Velodyne HDL 32-beam rotating LiDAR scanner, an IMU, and a GPS/GNSS receiver. Furthermore, we provide Graph-SLAM poses [37]. In total, the dataset comprises 15 highly diverse and challenging urban scenes. The data collection runs cover a wide range of illumination conditions, weather conditions, and structural diversity. Figure 3 illustrates exemplary RGB images and the recording locations. The dataset was recorded over the course of multiple years in the city of Freiburg, Germany. The dataset key statistics are listed in Tab. 1. To evaluate our approach, we manually annotated 50 ego-view RGB images from five data collection runs not included in the training set.

Table 2: Model performances when trained on the *Freiburg Pedestrian Scenes* / Vistas datasets and evaluated on the *Freiburg Pedestrian Scenes* dataset. We denote the IoU values in %.

| Annotation Source | ■ Road | ■ Pedestrian | ■ Crossing | ■ Obstacle | Mean |
|---|---|---|---|---|---|
| Mapillary Vistas [7] | 12.1 | 20.0 | 0.5 | **89.2** | 30.4 |
| Ego | 0 | 37.3 | 0 | 85.8 | 30.8 |
| Ego + Tracklets | 35.9 | 67.5 | 43.4 | 88.3 | 58.8 |
| Map Reprojection | **38.4** | **69.2** | **48.2** | 85.9 | **60.4** |

We also hand-annotated large sections of the traversed areas with a semantic BEV map in order to be able to compare aggregated and ground-truth maps. Exemplary visualizations of this map are visualized in Fig. 5.

# 5    Experimental Results

We compare our automatic annotation approach with several baseline approaches. All results are listed in Tab 2. We first evaluate a model trained on the Vistas dataset [7]. To perform a quantitative comparison, we re-map the Vistas classes to the *Freiburg Pedestrian Scenes* class labels. While we obtain a high test mean Intersection-over-Union (mIoU) of $62.9\%$ on the Vistas test split, we obtain relatively low mIoU values of

Table 1: *Freiburg Pedestrian Scenes* dataset details

| Modality | Quantity | Frequency [Hz] | |
|---|---|---|---|
| Stereo RGB | 260k | 5 | |
| LiDAR | 490k | 9 | |
| IMU | 4.2M | 100 | |
| GPS | 490k | 9 | |
| SLAM poses | 49k | 1 | |
| Map annotations | $112523 \, \mathrm{m}^2$ | - | |

$30.4\%$ on our dataset. This can be attributed to the domain gap between the two datasets due to the inconsistent camera viewpoints. Leveraging the robot ego-trajectory yields greatly improved results for the *Pedestrian* class but cannot account for any other semantic class. Our tracklet-based method, in contrast, shows better performance than the baseline model in all classes but the *Obstacle* class. What's more, our experiments indicate an improvement of IoU values when leveraging the aggregated semantic surface map (constituting dataset $\mathcal{D}_1$) for model training. This is most likely due to the larger number of annotated pixels and the increased annotation consistency due to prediction aggregation. We illustrate qualitative results in Fig. 4. Generally speaking, the model trained on Vistas shows many false-positive *Road*-classifications due to the camera viewpoint bias present in the Vistas dataset. We also observe that segmentation masks of our best-performing model are well aligned with the ground-truth annotations. However, due to the challenging visual similarity between ground classes, not all areas are predicted correctly. Most incorrect predictions are produced in crossing regions and in places where sidewalks and streets are not easily distinguishable (see Fig. 4, failure cases).

## 5.1    Evaluation of Semantic Surface Maps

We qualitatively evaluate the semantic surface maps obtained with our approach. To generate the maps, we use our segmentation model and aggregate its predictions as described in Subsec. 3.2. Figure 5 illustrates maps produced with our approach and the respective ground truth maps. We observe that the generated maps exhibit more consistent class assignments compared to the ego-view image predictions due to the prediction aggregation procedure for map generation. We observe that in most areas, the predicted ground class equals the actual ground class. In particular, the classes *Pedestrian* and *Road* align well with the ground truth areas. Challenging street crossings are accounted for in all regions. However, we note that the spatial extent of some crossing regions due to *Crossing/Road* and *Crossing/Pedestrian* misclassifications leaves further room for improvement. For more experimental evaluations, please refer to Suppl. Material Sec. H and I.

## 5.2    Limitations

Despite the fact that the aggregated maps are mostly well-aligned with the ground truth maps, not all annotations are correct, leading to partial bleeding of classes into each other. The class *Crossing* is particularly challenging for two reasons: Firstly, the annotations produced by our approach are not always consistent since not all street crossings are covered by observed trajectories. Secondly,

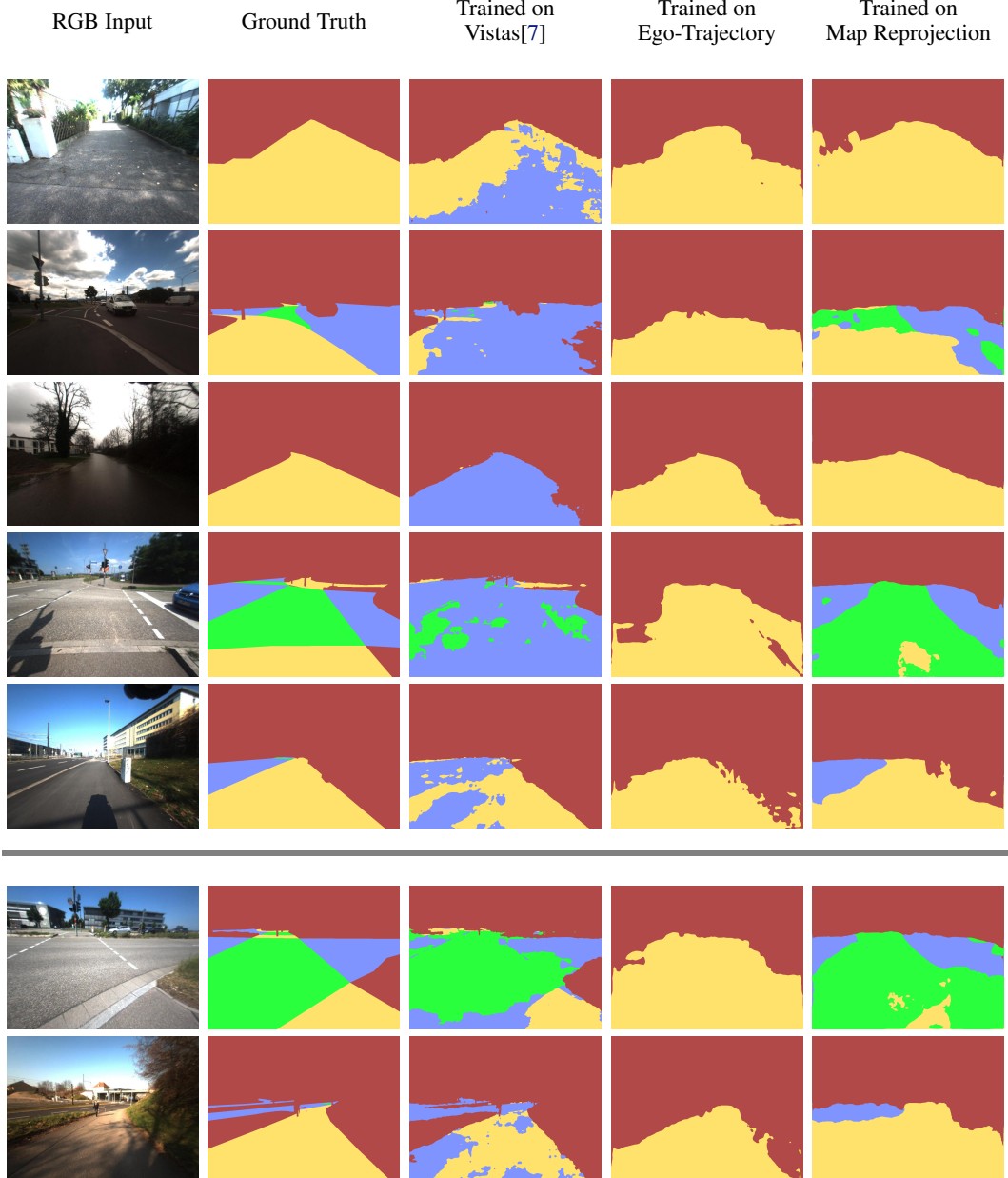

Figure 4: Qualitative results of models trained on different datasets and evaluated on the test split of our *Freiburg Pedestrian Scenes* dataset. We visualize RGB input images, ground truth annotations, and model predictions. Images below the horizontal gray line illustrate failure cases. Color-codes for the semantic classes are: ■ *Road*, ■ *Pedestrian*, ■ *Crossing*, ■ *Obstacle*.

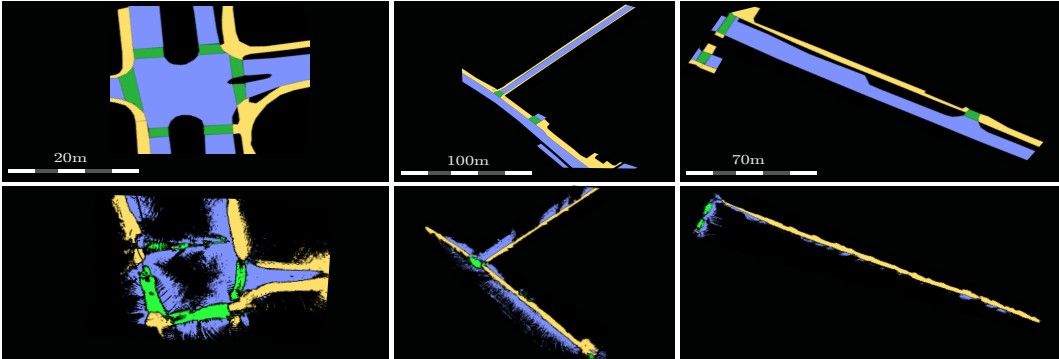

Figure 5: Exemplary illustrations of maps produced with our approach (best viewed zoomed in). The top row shows the respective ground-truth map and the bottom row shows the aligned semantic surface maps obtained with our aggregation approach. Color-codes for the semantic classes are identical to Fig. 4. Black color denotes not annotated / unobserved areas.

the *Crossing* pixels have substantial overlap in terms of texture with pixels of classes *Road* and *Pedestrian*, requiring the model to rely on contextual information such as line markings, which is not present in all scenes. Furthermore, our approach requires highly accurate localization, sensor calibration, and object tracker performance in order to generate correct annotations. Finally, the annotation quality depends on the behavior of traffic in accordance with traffic rules. If pedestrians jaywalk to cross streets or vehicles drive in pedestrian areas, the annotations can be inconsistent, leading to reduced model performance.

# 6 Conclusion

In this work, we showed how a semantic segmentation model for urban surface segmentation can be trained from projections of the ego-trajectory and projections of tracklets of other traffic participants. We also showed that the segmentation model can be further improved via self-distillation by spatially aggregating the model predictions into a semantic map. Regarding possible future work, there is room for improvement in terms of overall segmentation quality. Furthermore, future work might include the extension of the approach to more types of traffic participants such as bicycles and railways to accommodate more urban environments and annotating higher-level map attributes compared to surface types such as road graphs and lane graphs.

### Acknowledgments

We would like to thank our former colleagues at AIS for providing parts of the datasets. This includes in particular Noha Radwan, .... We would also like to thank the reviewers for their helpful comments and suggestions. Finally we would like to thank the DFG for providing funding for the project under grant DFG BU 865/10-2 - Autonomous Street Crossing with Pedestrian Assistant Robots.

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
