# OpenReview forum: "TrackletMapper: Ground Surface Segmentation and Mapping from Traffic Participant Trajectories"
_robot-learning.org/CoRL/2022/Conference — CoRL 2022 Poster_

### Official Review · Reviewer_x2BQ · 2022-07-22

**Originality:** Very Good
**Technical Quality:** Excellent
**Clarity Of Presentation:** Very Good
**Impact:** 3

**Recommendation:**

Strong Accept: I recommend accepting the paper and will argue for my recommendation even if other reviewers hold a different opinion.

**Summary:**

The paper proposes a framework for ground surface segmentation and mapping in RGB images. The main motivation is to facilitate the labeling effort for training learning-based ground segmentation methods. The proposed approach does not require any human labeling and infers the ground classes sidewalk, road, street crossing, and obstacles from projected ego-motion and object tracklets estimated with an object detection and tracking approach. To increase the number of annotations, a ground segmentation network is trained using this data and its predictions are used in a second iteration to generate a semantic map. The authors provide a dataset with ground segmentation labels estimated by their method as well as a labeled test set. The experiments show that training with additional labels generated from object tracklets and with the surface model increases the segmentation performance on the test set.

**Issues:**

Minor issues:

- It is not clear to me how the approach selects tracklets to be projected into the camera image. If trajectories are occluded by other objects or just very far away, how are these excluded from the projection? Looking at Fig 1., it seems like the past trajectory of both pedestrians on the right is not projected into the image since is occluded by them. How is this achieved?

- Are the surface map-based annotations in D1 solely extracted from the aggregated map? If yes, I wonder if they could have been combined with the previously generated ego-motion and tracklet-based annotations? This could increase the robustness in case wrong semantic predictions lead to a corrupted aggregated map.

- How are the surface patches in Section 3.2 extracted?

- Since a surface patch contains multiple pixels, what if the per-pixel predictions differ? How is the belief h in Equation (3) computed for a surface patch given contradicting pixel-wise predictions?

- The caption in Table 2 suggests that all models are trained on the proposed dataset. Is this a mistake, since the Mapillary Vistas is a different annotation source?

- Do the authors have an explanation for why the map reprojection decreases the segmentation performance for obstacles? Are the obstacles derived from bounding boxes as mentioned in Supplementary D also included in the map reprojection?

- It would be very nice to see the corresponding annotation results for the images in Figure 4. I could imagine that the human-annotated ground truth masks differ from the automatically generated supervision signals, therefore it is harder to assess how good the corresponding prediction is.

**Quality Of The Limitations Section:**

Limitations are addressed clearly

**Reviewer Expertise:**

4: The reviewer is confident but not absolutely certain that the evaluation is correct

**Robotics Focus:**

Sufficient demonstration on hardware

**Strengths And Weaknesses:**

Strengths:
- The paper is very well written and easy to follow.
- The experimental section supports the claim that supervision signals from object tracklets boost the ground segmentation performance.
- The authors record a diverse dataset and plan to release it along with the code which will be beneficial for the community.
- The limitations are well discussed.

Weaknesses:
- There are just some minor unclarities that are listed in the Issues section.

**Summary Of Recommendation:**

The paper is very well written and technically sound. The authors clearly motivate the problem and describe their annotation procedure in detail. I acknowledge the recording and annotation of a public dataset which makes it possible to assess the quality of the annotation when using them for training a segmentation network. The experimental section shows that using tracked objects and an aggregated surface map leading to more annotation labels improves the training of the segmentation network. There are just a few unclarities listed below that should be discussed and possibly revised in the manuscript.

---

> ### Author Response · Authors · 2022-08-24
> **Thank you for your valuable comments (1/2)**
>
> We thank the reviewer for taking the time and effort to provide a thorough and thoughtful review. Please find below our responses to the questions asked in the review.
>
> > Q1: It is not clear to me how the approach selects tracklets to be projected into the camera image. If trajectories are occluded by other objects or just very far away, how are these excluded from the projection? Looking at Fig 1., it seems like the past trajectory of both pedestrians on the right is not projected into the image since is occluded by them. How is this achieved?
>
> We thank the reviewer for asking this relevant question. Firstly, not all observed trajectories are actually used for annotation as far away trajectories suffer to a larger extent from projection inaccuracies if any of the involved calibrations (camera-lidar time-synchronization, lidar extrinsic calibration, camera intrinsic/extrinsic calibration) is not accurate. In order to handle occlusions well, after projecting the tracklets into the current camera frame, we use the LiDAR point cloud to detect any objects between the tracklet and the robot camera. All pixels that are marked as obstacles this way “overwrite” the tracklet pixels as obstacles. This helps make the annotations much more consistent as tracklet occlusions regularly occur in the busy scenes that were present during data collection runs.
>
> > Q2: Are the surface map-based annotations in D1 solely extracted from the aggregated map? If yes, I wonder if they could have been combined with the previously generated ego-motion and tracklet-based annotations? This could increase the robustness in case wrong semantic predictions lead to a corrupted aggregated map.
>
> We thank the reviewer for asking this intriguing question. We currently combine D1 with the observed tracklets in one fashion: We use the (compared to D0) larger annotated ground surfaces for the classes “Road” and “Pedestrian” with the tracklets for vehicles and pedestrians. This allows us to label even more pixels as “Crossing” as the union of these sets is much larger compared to using the union of tracklet-based annotations alone. However, we agree with the reviewer that in principle the annotations could be used further to improve the aggregated surface map, in particular in regions where the belief vector is multi-modal (two or more classes are equally likely at a given surface patch). Albeit we did not conduct any experiments to investigate this further.
>
> > Q3: How are the surface patches in Section 3.2 extracted?
>
> Surface patches are implemented as a 2D square grid. For our experiments, we use a grid resolution of 5cm x 5cm. Please note that the corners of each grid cell are also assigned a z-height value (vertical distance from the map coordinate frame origin). Therefore, the surface patch might not look straight into the sky but might be tilted in regions where non-level terrain is observed.
>
> > Q4: Since a surface patch contains multiple pixels, what if the per-pixel predictions differ? How is the belief h in Equation (3) computed for a surface patch given contradicting pixel-wise predictions?
>
> A given surface patch aggregates all class predictions falling into its region coming from any camera images in which the patch is visible. This means that almost certainly, the model predictions differ and are not consistent (typical behavior for any semantic segmentation model). Our proposition is that we aggregate class predictions by updating the class-belief for each surface patch whenever it is observed in the camera image. The belief is a vector of likelihoods for each class. In an extreme case, this might mean that the (normalized) belief vector might look like [0.49, 0, 0, 0.51, 0]. This means that after aggregations, the class likelihood for class 0 and 3 is almost equal, indicating high model insecurity regarding the class of that patch. Using the argmax function, we would set the class of this patch to 3. In practice, most belief vectors are less ambiguous, for instance [0.10, 0.15, 0.60, 0.10, 0.05]
>
> > Q5: The caption in Table 2 suggests that all models are trained on the proposed dataset. Is this a mistake, since the Mapillary Vistas is a different annotation source?
>
> We thank the reviewer for pointing out this inaccuracy. We corrected the table caption.

---

> > ### Author Response · Authors · 2022-08-24
> > **Thank you for your valuable comments (2/2)**
> >
> > > Q6: Do the authors have an explanation for why the map reprojection decreases the segmentation performance for obstacles? Are the obstacles derived from bounding boxes as mentioned in Supplementary D also included in the map projection?
> >
> > We are not entirely sure what effect causes this result. Obstacles are not included in the map projection used as dataset D1. We only project ground-surface classes back into the RGB images. Obstacles are then derived from the bounding boxes and image stixels the same way we obtain them for dataset D0 (tracklet-only based). One possible explanation for the slightly decreased obstacle IoU value is the change in the class annotation distribution. The reprojection-based annotation procedure (D1) produces annotations for more pixels, possibly leading to slightly more inconsistent labels for the class obstacle (bleeding of annotations into obstacles that are not detected by the object detector and thus incorrectly not “overwritten” as being obstacles).
> >
> > > Q7: It would be very nice to see the corresponding annotation results for the images in Figure 4. I could imagine that the human-annotated ground truth masks differ from the automatically generated supervision signals, therefore it is harder to assess how good the corresponding prediction is.
> >
> > We thank the reviewer for their idea. We would like to note that we mainly used data collection runs for the test split where the SLAM solution did not converge properly. Therefore, we do not have automatically generated annotations using our approach for the images from the test split of our dataset. If the reviewer is interested in assessing the quality of the automatically generated annotations using our approach, we would like to refer them to Fig. 8 and 9 in the supplementary material, where we visualize many annotations.

---

> > > ### Comment · Reviewer_x2BQ · 2022-08-26
> > > **Response**
> > >
> > > Thank you for the clarifications. Is there a revised version of the paper indicating your changes? If so, I could either not find it or it was not uploaded.

---

> > > > ### Author Response · Authors · 2022-08-26
> > > > **Response**
> > > >
> > > > We thank the reviewer for pointing out the missing color-highlighted updated version of the manuscript. The manuscript was updated: https://openreview.net/forum?id=qr0wqg8NqkL&noteId=k6BWBfQy9i

---

### Official Review · Reviewer_sqbj · 2022-07-27

**Originality:** Very Good
**Technical Quality:** Good
**Clarity Of Presentation:** Good
**Impact:** 3

**Recommendation:**

Strong Accept: I recommend accepting the paper and will argue for my recommendation even if other reviewers hold a different opinion.

**Summary:**

In this paper, the authors propose a method to automatically label the roads according to their possible traversability by vehicles, pedestrians or none. They use the tracklets of vehicles, pedestrians and segmentation results of DeeplabV3+ for this purpose. My comments are as follows.

**Issues:**

- The number of revisits and its effect on the improvement of the segmentation should be discussed.
- The possibility of a SLAM algorithm for creating a semantic map for the collected data should be discussed.

**Quality Of The Limitations Section:**

Limitations are addressed clearly

**Reviewer Expertise:**

5: The reviewer is absolutely certain that the evaluation is correct and very familiar with the relevant literature

**Robotics Focus:**

Highly relevant to robotics but no hardware experiments

**Strengths And Weaknesses:**

Is the surface considered to be flat? What would happen if there were slopes/irregularities on the road surface?

Do you perform a manual post-process to check the correctness of the final labels such as tracklet continuation, etc.?

If an HD map was possible, would this increase the accuracy?

It can be interesting if you can report all the results of the trained mIoU's on related datasets for Tab.2.

How long did it take approximately to label a ground truth image?

Did you consider multi-class labels for pixels that may belong to more than one class?

Did the robot platform visit the same locations more than once? If so, did it change/improve the labeling of the road? This point should be discussed further.

**Summary Of Recommendation:**

Segmentation of the roads according to their traversability is a hard work, both from the point of view of data collection and platform maintenance. The limitations are also clearly addressed. If the data can be made public, it will be good addition to the community.

---

> ### Author Response · Authors · 2022-08-24
> **Thank you for your valuable comments**
>
> We thank the reviewer for taking the time and effort to provide a thorough and thoughtful review. Please find below our responses to the questions asked in the review.
>
> > Q1: Is the surface considered to be flat? What would happen if there were slopes/irregularities on the road surface?
>
> We do not assume a flat ground surface. Our ground surface mesh can have inclined surfaces as long as they are observable from our LiDAR point cloud. We added an additional figure (Fig. 5) in the supplementary material to visualize an exemplary ground surface that features inclined surfaces.
>
> > Q2: Do you perform a manual post-process to check the correctness of the final labels such as tracklet continuation, etc.?
>
> We thank the reviewer for asking this relevant question. We do not leverage any post-processing scheme for the tracklets. We also do not manually remove any samples or maps from the dataset before training. The only manual step we perform is checking the validity of the SLAM solutions. For some data collection runs, the SLAM solution did not converge well and led to inconsistent ego- and tracklet-trajectories. We removed these data collection runs from the dataset we intend to publish.
>
> > Q3: If an HD map was possible, would this increase the accuracy?
>
> The authors are not sure whether they understood the question correctly. If the reviewer refers to the possibility of projecting a manually obtained HD map into the RGB images, we expect the segmentation model to exhibit even stronger performance (if localization is accurate) since the manually annotated map will be more accurate compared to the aggregated map used for our approach. The intent of our approach is to not rely on HD maps but rather learn semantic surface maps facilitating navigation tasks. Please let us know if this was not your question.
>
> > Q4: It can be interesting if you can report all the results of the trained mIoU's on related datasets for Tab.2.
>
> We would like to inquire further about this. Does the reviewer mean that the model trained on our own datasets D0 and D1 could be evaluated on semantic segmentation datasets such as CityScapes and Vistas? Or does the reviewer mean that the model should rather be trained on CityScapes (in addition to Vistas) and evaluated on our dataset to further assess the significance of the domain gap between the camera viewpoints?
>
> > Q5: How long did it take approximately to label a ground truth image?
>
> It took us approx. 15 minutes to manually annotate a single image with the ground-truth annotations.
>
> > Q6: Did you consider multi-class labels for pixels that may belong to more than one class?
>
> We thank the reviewer for asking this really interesting question. We played around with the idea of using the output of the softmax (manuscript, line 153) as a per-pixel class label instead of the argmax of this vector. We agree that this information might contain more relevant information than the most likely class alone (which is the output of the argmax on the belief vector) and would be able to assess the model uncertainty for surface patches when multiple classes are equally likely. However, in the end, we did not experiment further with this idea. We agree that this information might be helpful for training a better-performing model and will be considered for follow-up work.
>
> > Q7: Did the robot platform visit the same locations more than once? If so, did it change/improve the labeling of the road? This point should be discussed further.
>
> We thank the reviewer for this relevant comment. For multiple datasets, the robot visits the same place more than once. Usually, this comes from the fact that the robot visits the same streets going forth and coming back. However, this also means that the camera viewpoint will not be the same in both instances. Visiting the same location twice also adds the benefit of introducing SLAM loop closures which are helpful for obtaining accurate SLAM poses. In general, multiple passes over the same area are useful as they usually increase the number of labeled pixels (both based on the ego-trajectory and based on the observed trajectories).

---

### Official Review · Reviewer_vAHU · 2022-07-31

**Originality:** Poor
**Technical Quality:** Poor
**Clarity Of Presentation:** Good
**Impact:** 2

**Recommendation:**

Strong Reject: I recommend rejecting the paper and will argue for my recommendation even if other reviewers hold a different opinion.

**Summary:**

This paper focuses on an automated data labeling pipeline where camera poses are found, per image detection and tracking is done and then everything is fused to the world frame. The labeling is aggregated in the world frame and projected back to each image. They use the pipeline to annotate images and train a semantic segmentation model with deeplab.

**Issues:**

See Weaknesses.

**Quality Of The Limitations Section:**

Limitations section not present

**Reviewer Expertise:**

5: The reviewer is absolutely certain that the evaluation is correct and very familiar with the relevant literature

**Robotics Focus:**

Irrelevant to robotics

**Strengths And Weaknesses:**

Weaknesses:
The idea of labeling images in 3D and projecting them back to the images is not new and has been done before: https://arxiv.org/pdf/1702.07836.pdf
The paper does not present a novel idea and rather uses bits and pieces of prior work.


**Summary Of Recommendation:**

The paper is more like a systems paper and does not present new perspective/insight.

Post Rebuttal: I still think the paper is very incremental. It is not clear what this dataset provides beyond all the other large scale datasets such as the ones released by Waymo and Lyft. Having a paper about how to collect new dataset without really justifying what is wrong with other datasets is not a good motivation. In order for this dataset to be on the next level, it should have significantly more diversity in locations and size of dataset.

---

> ### Author Response · Authors · 2022-08-24
> **Thank you for your valuable comments**
>
> We thank the reviewer for drawing the comparison with the linked publication which will in the following be denoted as “P”. While we agree with the reviewer that the paper visits vaguely similar ideas on a fundamental level, we argue that both the approach and the application differ substantially and do not justify the claims made in the review. However, we would like to further address your concern. Please provide any additional feedback about how we can improve this (see discussion below).
>
> We summarize P as follows: The paper proposes a novel method of synthetically placing objects in RGB images of indoor scenes to produce new and augmented RGB images. Objects are placed according to high-level scene properties such as estimated support surfaces in order to find regions fit for object placement. Furthermore, depth estimation is used to determine the correct size of the objects given their distance from the camera and their physical real dimensions. These physically realistic and automatically obtainable synthetic composite training images can be used to create large annotated datasets and can boost the performance of object detectors.
> In the following, we expand on the significant differences between our work and P:
>
> - P proposes to create synthetic composite RGB images along with annotations for the training of object detectors. We do not create synthetic RGB images (we never alter the originally recorded RGB images) but instead use object tracklets as an annotation source.
> - P does not use tracklets of moving objects to derive a supervisory signal for a downstream learning-based model, in contrast to our work.
> - P does not use LiDAR, Camera poses, SLAM, or any kind of robotic system (in contrast to our work).
> - P evaluate their method for the task of indoor object detection. We tackle the task of outdoor semantic segmentation and mapping (a different task and a different type of environment).
> - P does not model self-distillation as a means to improve model predictions. Neither do they aggregate semantic maps or other spatial structures to self-distill their model for performance.
>
> Therefore, we argue that both the application and the idea proposed in P differ substantially from the ideas and applications in our manuscript, ultimately not justifying the evaluation by the reviewer.
>
> Please let us know if these comments address your concerns.
>
> Finally, we would like to mention that the limitations section, in contrast to the reviewer’s comments, is present.

---

### Official Review · Reviewer_a8Do · 2022-08-01

**Originality:** Good
**Technical Quality:** Very Good
**Clarity Of Presentation:** Very Good
**Impact:** 2

**Recommendation:**

Weak Reject: I recommend rejecting the paper, but will not argue for my recommendation if the majority of other reviewers have a different opinion.

**Summary:**

This paper presents a two-stage approach to automatically label ground surface segmentation (roads, pedestrian, crossing, obstacle). In the first stage, the paper leverages pedestrian and vehicle tracking as well as the robot's trajectories to produce sparse, trajectory-based annotations. In the second stage, the page leverages a learning-based segmentation model to dilate the labels to fill more pixels. Using this automatic annotation pipeline, the paper also introduces an annotated dataset.

**Issues:**

- Figure 1: why are there no obstacle labels on the top-left area of the image?
- The author's definition of 'crossing' can be more abstract. Semantically, crossing are specified patch of areas where the pedestrians and vehicles both may appear on it. Defining it as intersections of 'pedestrian' and 'road' may result in noisy labelling.
- The authors frequently mention 'model performance'. What is this performance? What metric are the authors using to define this 'performance'?
- $^W\bf{T}_B$ and $^B\bf{T}_C$ look confusing when put into the equations. Consider $\bf{T}^W_B$ and $\bf{T}^B_C$
- Unclear what surface patches are defined in section 3.2
- It is very confusing how 'unclear' is handled in section 3.3. Consider refining this part.
- In section 5.1, "We observe that in most areas, the predicted ground class equals the actual ground class." I don't think this is the case. Can you show this with better illustrations? Maybe define some metric and conduct some quantitative evaluation (such as IoU).

**Quality Of The Limitations Section:**

Additional details required

**Reviewer Expertise:**

4: The reviewer is confident but not absolutely certain that the evaluation is correct

**Robotics Focus:**

Sufficient demonstration on hardware

**Strengths And Weaknesses:**

The paper overall is clearly written with its approach and evaluation clearly documented.

I am very excited about the paper's proposed self-distillation procedures. This is a great idea for weak supervision. The presence of pedestrians and vehicles on the roads are light in most environments and their behavior can be somewhat stochastic, but segmentation labels contain very rich information. These procedures can be promising in filling the gaps among the sparse annotations to achieve the same level of information richness as full segmentation annotations much like filling in the missing pixels of an image.

I also like the evaluation performed in comparison with the Vistas dataset. It really shows the domain gap between datasets collected from different perspectives. The dataset also appears to be very large in scale.

Unfortunately, I think the positioning of the paper is a bit off or unfinished. This paper aims to produce labels for a dataset. Although from the experiments the paper shows that its dataset is more suitable for pedestrian-oriented tasks than autonomous driving datasets, the generated labels are still far from perfect as seen from figures 4, 5 and the video in the supplementary materials. Perhaps a good question to ask is: what task do the authors think this dataset is useful for? If it is for navigation, this dataset can be dangerous to rely on. For example, I observe ground surfaces that should be roads or even obstacles to be labelled as pedestrian and we certainly don't want to see a robot wandering in those places. I also observe 'holes' in areas that are predominantly 'pedestrian' or 'road'. The paper's proposed method outputs labels that already have a discounted quality from the human-labelled data. The models that are trained on the paper's dataset will surely only perform worse in their respective tasks than the models trained on more accurately labelled datasets.

Therefore, I think the paper is unfinished and has limited impact in its current stage. It would be very helpful if the authors can identify a task that may benefit from this dataset (or maybe just the segmentation task). Despite the imperfect labels, the dataset is very large in scale. If the scale of the dataset can compensate the inaccuracies in labels and the authors can demonstrate this in an experiment. This would highlight the utility of their dataset.

Alternatively, I see this as an opportunity for a learning-based robot navigation task. The outputted segmentation maps, though dangerous to rely on, can still be useful in providing contextual information for certain downstream robot navigation tasks. If the authors can integrate their approach in a navigation model and demonstrate its success, I think it will be a strong paper.

**Summary Of Recommendation:**

This paper has limited impact because the quality of the automatically produced labels are questionable and more experiments are needed to support the utility of the dataset. It is unclear what downstream tasks the authors have in mind that would benefit from this dataset. It is also unclear whether the scale of the dataset might help in reducing the negative impact of the low-quality labels. There is novelty in the paper's approaches, so I rate this paper a weak reject.

---

> ### Author Response · Authors · 2022-08-24
> **Thank you for your valuable comments (1/2)**
>
> We thank the reviewer for taking the time and effort to provide a thorough and thoughtful review. Please find below our responses to the questions asked in the review.
>
> > Comment 1: Unfortunately, I think the positioning of the paper is a bit off or unfinished. This paper aims to produce labels for a dataset. Although from the experiments the paper shows that its dataset is more suitable for pedestrian-oriented tasks than autonomous driving datasets, the generated labels are still far from perfect as seen from figures 4, 5 and the video in the supplementary materials. Perhaps a good question to ask is: what task do the authors think this dataset is useful for? [...] The models that are trained on the paper's dataset will surely only perform worse in their respective tasks than the models trained on more accurately labelled datasets.
>
> We thank the reviewer for critically assessing the utility of the approach. As discussed in the Introduction of the manuscript, our intended use case for the approach is automatic ground surface segmentation and mapping. While we demonstrate the approach for a robot operating mainly in urban areas alongside pedestrians, the approach could generally also be leveraged to generate annotation data for autonomous vehicles, where the ego-vehicle trajectory indicates drivable surfaces for vehicles and not surfaces fit for pedestrian usage. Being able to find legit and safe trajectories for the robot based on the created maps is another focus of this work. To underline the sufficient quality of the generated maps, we conduct additional planning experiments, as described in the response below.
>
> We generally agree that the annotations obtained with our approach do not have the same quality as manual annotations. The advantage of the proposed approach, however, is its scalability. It is easy and inexpensive to gather more data in order to obtain more annotations, which should in general help the model to generalize better to unseen scenes. For example, on 10% of the data, our approach achieves a mIoU of 55.0% which increases to 60.4% when the entire dataset is used for training. Therefore, we are convinced that more (and even weak) annotations generally improve model performance. The advantage of our approach lies in the fact that it automatically generates and furthermore makes use of cheap imperfect annotations instead of relying on expensive although more accurate annotations.
>
>
> > Comment 2: Therefore, I think the paper is unfinished and has limited impact in its current stage. It would be very helpful if the authors can identify a task that may benefit from this dataset (or maybe just the segmentation task). Despite the imperfect labels, the dataset is very large in scale. [...]. If the authors can integrate their approach into a navigation model and demonstrate its success, I think it will be a strong paper.
>
> We thank the reviewer for their helpful comment.We believe that the approach proposed in this paper can be beneficial for several downstream robot navigation tasks including localization and path planning. To demonstrate this, we added experiments with an A* planner using the generated BEV semantic maps, which generates safe and efficient trajectories for the scenarios we evaluated. We added detailed explanations and visualizations to the supplementary material, Sec. I and Fig. 7.
>
> > Q1: Figure 1: why are there no obstacle labels on the top-left area of the image?
>
> We thank the reviewer for pointing out this inaccuracy. The authors omitted to mark these objects as they were deemed irrelevant for presenting the idea of the approach but we agree that the inconsistency might be confusing. Version 2 of the manuscript contains consistent labels of obstacles.
>
> > Q2: The author's definition of 'crossing' can be more abstract. Semantically, crossing are specified patch of areas where the pedestrians and vehicles both may appear on it. Defining it as intersections of 'pedestrian' and 'road' may result in noisy labelling.
>
> We thank the reviewer for raising this concern. We must differentiate between two definitions here. On one hand, the actual definition according to pedestrian crossing markings on the road or sections of road between lowered curbs on both sides. However, we cannot use this definition for providing automatic annotations according to our annotation scheme since we do not have access to paths, where vehicles/pedestrians “may eventually” appear. We only know where they actually appeared during data collection. Therefore, in terms of actual pixel annotations, we can only define “Crossing” where we actually know that both types of traffic participants covered the same patch of ground. We agree that this definition leads to incomplete or potentially noisy labels. This aspect was also discussed in the Limitations section of the manuscript.

---

> ### Author Response · Authors · 2022-08-24
> **Thank you for your valuable comments (2/2)**
>
> > Q3: The authors frequently mention 'model performance'. What is this performance? What metric are the authors using to define this 'performance'?
>
> We use the term performance in accordance with our performance metric which we define as the class-wise IoU value (e.g., Tab. 2).
>
> > Q4: Unclear what surface patches are defined in section 3.2
>
> We thank the reviewer for requesting this clarification. Surface patches are implemented as a 2D square grid. For our experiments, we use a grid resolution of 5cm x 5cm. Please note that the corners of each grid cell are also assigned a z-height value (vertical distance from the map coordinate frame origin). Therefore, the surface patch might not be horizontal but even tilted, for example,  in regions where non-planar or non-horizontal terrain is observed.
>
> > Q5: It is very confusing how 'unclear' is handled in section 3.3. Consider refining this part.
>
> Consider this rephrasing of the explanation: The annotation image is initialized with the class Unknown for all pixels. Now, we mark all pixels that contain observed tracklets with their respective class. Finally, we mark all obstacles if they are present. Now, all remaining pixels have the class Unknown associated with them as we obtained no information about them. In order to train a model with images annotated in this fashion, it is common practice to ignore all labels with the class Unknown. In terms of minimizing a cross-entropy loss this means that we simply do not add pixels to the loss term if their ground-truth label is Unkown (i.e. the loss weight for class Unknown is zero).
>
> > Q6: In section 5.1, "We observe that in most areas, the predicted ground class equals the actual ground class." I don't think this is the case. Can you show this with better illustrations? Maybe define some metric and conduct some quantitative evaluation (such as IoU).
>
> We thank the reviewer for making this suggestion. We added IoU, Precision, and Recall metrics for five exemplary BEV semantic maps in the supplementary material, section H. We performed a qualitative and quantitative evaluation of these maps and discussed limitations.

---

### Meta-Review · Area_Chair_21Re · 2022-08-12

**Recommendation:** Accept (Poster)
**Confidence:** 3

**Metareview:**

The paper proposes a method for annotating ground surface types such as sidewalks, roads, and street crossings, using a both the egocentric view of the robot and the projections of other traffic participants to provide sparse semantic annotations with which a ground segmentation model can be trained.

The paper's **strengths** are:
- the idea is interesting and the part of self-distillation is a promising way for handling sparsely annotated datasets;
- the evaluation results regarding the domain gap among datasets is insightful.

The paper's **weaknesses** are mainly:
- that the contributions seems to be fragmented and the overall method seems to not be complete;
- discussion of the contribution of this method compared to state-of-the-art;
- a sufficient connection with the robot-learning community is missing.

The authors should provide a more complete and improved version of their work, and adequately address the issues raised by the reviewers during the rebuttal.

**Post-rebuttal assessment:** After inspecting reviews and the paper, with the updated material [here](https://openreview.net/forum?id=qr0wqg8NqkL&noteId=0tIX7QHVSJB) -- please next time post updates on a general comment, reviewers were not able to discover it in chains of comments-- I believe that the paper address a challenging topic for semantic segmentation during navigation, and has interesting ideas for overcoming sparsely-annotated data via distillation. Overall, I think that the paper can be presented in a conference, given the overall improvements made by the authors.

**Best Paper Nomination:**

No

---

> ### Author Response · Authors · 2022-08-24
> **Thank you for your valuable comments**
>
> We thank the area chair for their critical assessment of the paper’s strengths and weaknesses. We hope that the updated manuscript, the extended supplementary material with additional experiments, and the detailed responses to each of the reviewer’s concerns helped better expand on the ideas presented in this paper and improve the overall quality of the manuscript and supplementary material.
>
> This includes but is not limited to a qualitative and quantitative experimental evaluation of the generated semantic surface maps, a planning experiment demonstrating the utility of the generated maps, and the addition of clarifications and extensions to the manuscript text.
>
> Finally, we hope that our comments regarding the criticism of redundancy with related work, raised by reviewer vAHU, helped clarify the fundamental differences between our work and the work mentioned by the reviewer.